# Latent Structure of Affective Representations in Large Language Models

## Abstract

The geometric structure of latent representations in large language models (LLMs) is an active area of research, driven in part by its implications for model transparency and AI safety. Existing literature has focused mainly on general geometric and topological properties of the learnt representations, but due to a lack of ground-truth latent geometry, validating the findings of such approaches is challenging. Emotion processing provides an intriguing testbed for probing representational geometry, as emotions exhibit both categorical organization and continuous affective dimensions, which are well-established in the psychology literature. Moreover, understanding such representations carries safety relevance. In this work, we investigate the latent structure of affective representations in LLMs using geometric data analysis tools. We present three main findings. First, we show that LLMs learn coherent latent representations of affective emotions that align with widely used valence–arousal models from psychology. Moreover, a comparison with latent structure in human brainwave (EEG) data suggests possible similarities. Second, we find that these representations exhibit nonlinear geometric structure that can nonetheless be well-approximated linearly, providing empirical support for the linear representation hypothesis commonly assumed in model transparency methods. Third, we demonstrate that the learned latent representation space can be leveraged to quantify uncertainty in emotion processing tasks. Our results are based on experiments with the GoEmotions corpus, which contains ∼58,000 text comments with manually annotated sentiment.

## 1 Introduction

Recent advances in mechanistic interpretability have illuminated how large language models (LLMs) internally represent high-level semantic features such as factual knowledge, syntax, and sentiment. A growing body of work investigates whether such features are geometrically organized in semantically plausible, useful ways (Chatterley et al., 2025; Skean et al., 2024; Tigges et al., 2023; Lee et al., 2025).

Emotions provide a particularly compelling domain for studying geometric representations in LLMs because their semantic organization has been extensively studied in the psychology and neuroscience literature. Emotions can be described both in terms of discrete categories (e.g., anger, joy, fear) and along continuous affective dimensions such as valence (positive–negative) and arousal (calm–excited); in particular, this two-dimensional valence-arousal model (Fig. 1) of emotion has proved particularly dominant in the psychology literature (Bradley & Lang, 1999; Bliss-Moreau et al., 2020; Maleki et al., 2023). Because these structural models are based in human cognition, they provide a natural benchmark for probing whether LLMs encode emotional information in comparable ways to humans. If LLM latent spaces reproduce aspects of these psychological and neuroscientific models, it may not only deepen our understanding of how LLMs internally organize sentiment but also suggest potential parallels with natural intelligence.

The present study is situated in an *affective* context. More specifically, the affective computing literature has explored how computational systems can recognize and interpret human emotions across modalities such as speech and text (Calvo & D'Mello, 2010; Huang et al., 2023; Picard, 2000). Inspired by this body of work, we design text-based emotion classification tasks to probe whether the LLM's learnt latent representations recover known categorical clusters; we hope to show, in line

with previous ideas that distributed embeddings can encode affective dimensions (Shah et al., 2022), whether continuous affective dimensions such as valence and arousal emerge as dominant axes of organization. In other words, rather than emphasizing the *generation* of sentiment-bearing text, our analyses center on how models respond to (and internally structure) sentiment-encoded *inputs*.

**Related Work**   Our work engages with two prominent theories on the internal geometry of LLM representations. The first, the linear representation hypothesis, posits that high-level concepts are encoded as simple linear directions in activation space. This view is supported by studies showing that features like sentiment polarity can be identified with linear probes (Park et al., 2023) and manipulated through causal interventions along a linear axis to steer model outputs (Nanda et al., 2023; Park et al., 2023; Tak & Gratch, 2024; Jin et al., 2024). Conversely, the manifold hypothesis suggests that representations form more complex, nonlinear structures. For instance, recent analyses show that internal representations of semantic categories can form simplex-like hierarchical structures (Park et al., 2024), while other work has broadly identified geometric and manifold-like structures in neural representations of language (Mamou et al., 2020; He et al., 2024). By finding evidence for nonlinear latent structure in line with established psychological models, our work helps inform the current debate on representation geometry.

**Summary of Contributions**   The main contributions of our work are as follows:

1. **Representational similarities with human affective models.** We show that Gemma-2-9B, Mistral-7B, as well as LLaMA-3-70B-Instruct develop coherent internal representations of affective emotion. These representations align with established valence–arousal models from psychology. We also find similarities with latent structure in human brainwave (EEG) data.
2. **Evidence for nonlinearity.** We find that the geometry of LLM emotion representations exhibits modest nonlinearity (in line with the parabolic curvature of valence-arousal space). While we find that affective emotion geometry is locally amenable to linear analysis, our evidence for nonlinear global structure suggests that a purely linear representation hypothesis is insufficient.
3. **Applications to uncertainty quantification.** We demonstrate that the geometry of these structured representation spaces can be exploited to quantify predictive uncertainty in emotion recognition tasks, illustrating both practical utility and interpretability gains.

## 2   BACKGROUND

**Constructing Latent Space Representations**   We will employ two manifold learning methods for recovering latent space representations of language model embeddings: classical multidimensional scaling (MDS) and Isometric Feature Mapping (Isomap).

*Classical MDS* (Torgerson, 1952) takes a dissimilarity matrix $D = [d_{ij}]$ and constructs the doubly centered squared-distance matrix $B = -\frac{1}{2}JD^{\circ 2}J$ with $J = I - \frac{1}{n}\mathbf{1}\mathbf{1}^\top$. An eigendecomposition $B = Q\Lambda Q^\top$ then yields coordinates $z_i \in \mathbb{R}^k$ from the top-$k$ eigenpairs, chosen so that pairwise Euclidean distances in the embedding approximate the original dissimilarities in $D$. This closed-form solution provides a linear Euclidean representation of the data.

*Isomap* (Tenenbaum, 1997) extends MDS to account for nonlinear structure by replacing the raw Euclidean distances $d_{ij}$ with estimates of geodesic distances along the data manifold. In practice, this is achieved by constructing a $k$-nearest neighbor ($k$NN) graph $G$ over the embeddings and computing approximate geodesic distances $\tilde{d}_{ij}$ as shortest-path distances on $G$; classical MDS is then applied to the geodesic distance matrix. By incorporating this local neighborhood structure, Isomap can capture curvature in the embedding space and reveal deviations from purely linear structure.

**Valence-Arousal Model of Emotion**   A widely used framework in psychology and affective science conceptualizes emotions along two continuous dimensions: *valence*, which captures the degree of pleasantness or unpleasantness, and *arousal*, which indexes physiological activation or intensity (Russell, 1980; Bliss-Moreau et al., 2020; Kim et al., 2020; Maleki et al., 2023). Early work in this space originally posited circumplex emotion layouts centered around neutral (Russell, 1980). In recent years, however, psychology literature has increasingly adopted the notion of a parabolic (i.e., "V"-shaped) geometric layout of emotion (Kim et al., 2020; Maleki et al., 2023), owing to a

general correlation between valence intensity and arousal arising in the typical distribution of human emotion. Geometric visualizations of the dual-axis valence-arousal emotion layout are shown in Figure 1.

# 3 METHODS

## 3.1 PRELIMINARIES

To study the organization of emotional representations in LLMs, we begin by defining a notion of similarity between embeddings corresponding to different emotions. For each emotion pair $(i, j)$, we train a pairwise logistic regression classifier with L2 regularization on mean-pooled hidden state activations (see Section 3.2). We treat the test accuracy $\text{acc}_{ij}$ of this classifier as a proxy for the distance between emotions $i$ and $j$, with higher accuracy indicating greater separability. We then define by default $D_{ij} = \text{acc}_{ij}$. This yields a symmetric dissimilarity matrix that captures inter-emotion relationships in LLM activation space, since higher separation accuracy means the two emotions form well-separated activation clusters, while lower accuracy reflects overlapping representations. We also verified that our results are robust under alternative definitions of dissimilarity, such as activation cosine similarity (see Apx. A), distances to logistic regression hyperplanes, or alternative accuracy-to-distance transformations. Our logistic regression approach, however, is used by default as it provides a well-defined class-separating hyperplane, which plays a key role in our downstream uncertainty quantification application (see Sec. 6).

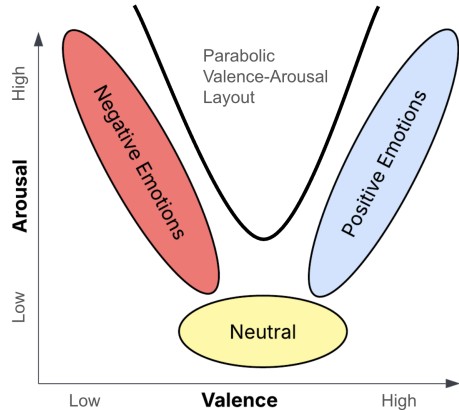

Figure 1: Parabolic valence–arousal model of affective space, based on Maleki et al. (2023).

## 3.2 METHODOLOGY

The main aim of our exploratory analysis is investigating the following hypothesis:

> *Do LLMs develop coherent internal representations of emotions that align with the valence-arousal model?*

To test this hypothesis, we conduct an activation-based probing analysis across Mistral-7B and Gemma-2-9B centered around the geometry of representations in an explicit emotion-classification setting. In line with affective computing practices, we feed each prompt to the target LLM and prompt the model to classify the emotion from the randomized list of GoEmotions choices, subsequently storing activations corresponding to both correct and incorrect classifications. All inference was performed in zero-shot mode, with no additional fine-tuning on GoEmotions or related datasets. For each text sample, we extracted mean-pooled hidden state activations from all transformer layers using forward hooks. Specifically, we collected the raw hidden states from each of the 32 layers in Mistral-7B and 42 layers in Gemma-2-9B, where each layer produces activations of dimension [1, sequence_length, hidden_size] since we process one sample at a time. We then applied mean pooling across the sequence dimension to obtain a single vector representation per layer for each model; we use layer outputs as opposed to sub-layer components, as in Ju et al. (2024) and Li et al. (2024). The activation vectors from correctly classified samples were used to train *pairwise* classifiers between emotions, enabling us to identify how the geometric organization of emotional representations changes across depth and to assess where representations are most discriminative. The incorrect activations were stored for future downstream analysis (see Section 6).

We focused on binary emotion comparisons rather than multiclass setups. This approach likewise avoids confounds from highly imbalanced class sizes (per-class correct sample counts range from 101 to 2,212 in Gemma-2-9B and 103 to 1,293 in Mistral-7B), yields clearer and more stable decision boundaries in activation space, and dovetails with MDS, which operates on pairwise dis-

similarities. For each emotion pair,[1] we balanced the dataset by downsampling the majority class, then trained a logistic regression classifier with L2 regularization using an 80:20 train-test split. The resulting classification accuracies were subsequently converted into dissimilarity values $D_{ij}$ as described in Section 3.1, providing the pairwise distances for downstream MDS analysis.

# 4 EXPLORATORY DATA ANALYSIS

## 4.1 EXPERIMENTAL SETTING

**Models** We examine two state-of-the-art transformer-based LLMs: Gemma-2-9B (Team et al., 2024) and Mistral-7B (Jiang et al., 2023). Gemma-2-9B is a 9-billion parameter, 42-layer model developed by Google DeepMind, while Mistral-7B is a 7-billion parameter, 32-layer model from Mistral AI. Although the exact training mixtures for these models have not been publicly disclosed, both were trained on large-scale, diverse web corpora that likely included sentiment-laden text, providing them with natural exposure to affective language. We also experimented with analyses on LLMs from the Qwen and LLaMA families but found that these models did not satisfy requisite sentiment recognition baselines necessary for our downstream experiments; further discussion is included in Apx. B. In addition, to assess generalization across both scale and training regime, we conducted a full replication of our pipeline on LLaMA-3-70B-Instruct, a substantially larger and instruction-tuned model (see Section 7).

**Datasets** Our primary dataset is *GoEmotions* (Demszky et al., 2020), a manually annotated corpus of 58,009 English Reddit comments labeled for 27 fine-grained emotion categories plus *Neutral*. The taxonomy comprises 12 positive, 11 negative, and 4 ambiguous categories, enabling analyses along both valence-aligned dimensions and discrete categories. Each comment was annotated by 3 or 5 independent raters (82 raters in total). We restrict our analyses to single-label examples exhibiting multi-rater agreement (∼83% of examples). As the largest manually annotated, fine-grained English emotion dataset to date, GoEmotions is a natural choice for our study.

As a secondary dataset to probe parallels between text-derived and neural emotion representations, we also run experiments on *FACED* (Chen et al., 2023), a large, open EEG resource with recordings from 123 participants using 32 electrodes (10–20 system). Participants watched 28 emotion-eliciting video clips spanning nine categories—four positive (amusement, inspiration, joy, tenderness), four negative (anger, fear, disgust, sadness), and one neutral. We use FACED to test whether the geometric structure we observe in LLM embeddings also exhibits potential similarities with structure seen in neural EEG responses, thereby further investigating a potential link between affective artificial and natural cognition.

## 4.2 SEMANTIC ANALYSIS OF EMOTION REPRESENTATIONS

In our layer-by-layer activation analyses (see Apx. A for further details), we found that mean pairwise emotional separability generally rose toward the middle layers while exhibiting a slight drop at the final layers. Additionally, we found that Gemma-2-9B exhibited significantly higher activation separability compared to Mistral-7B, with mean test accuracies in the >0.9 range across all layers compared to ∼0.55 to ∼0.89 in Mistral-7B. This similarly aligns with Gemma-2-9B's greater LLM emotion recognition performance in the initial emotion classification setting, with an estimated ∼19.4% correct classification rate in Gemma-2-9B compared to ∼13.7% in Mistral-7B.

The latent representations obtained with classical MDS show that LLMs do indeed learn coherent representations that mirror logical semantic structure. Figure 2 shows the 2D projections of the classical MDS embeddings for Gemma-2-9B and Mistral-7B, respectively; both models exhibit clear semantic clustering, with positive emotions (joy, love, gratitude) clustering together in the upper-right quadrant and negative emotions (sadness, anger, disgust) grouping in the upper-left quadrant.

---

[1]To ensure sufficient dataset size, we conducted our analyses on emotion categories with at least 100 correct LLM classifications across the dataset, resulting in 20 emotions for Gemma-2-9B and 16 emotions for Mistral-7B (spanning positive, negative, ambiguous, and neutral valences).

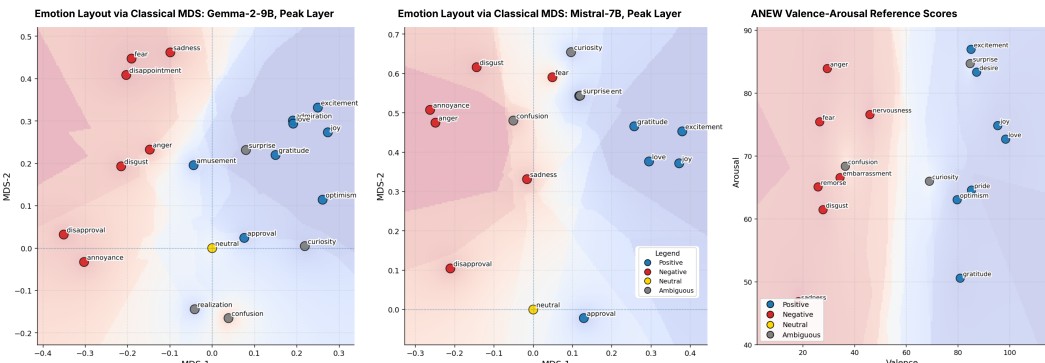

Figure 2: 2D classical MDS embeddings of internal latent emotion representations in Gemma-2-9B (left) and Mistral-7B (middle), as well as ANEW reference valence-arousal scores (right). Layers chosen correspond to peak separability (Gemma-2-9B Layer 20, Mistral-7B Layer 27) and are representative of overall patterns observed throughout. Both plots are anchored to fixed orientations with neutral at the origin; colors correspond to the given GoEmotions taxonomy, with $k$NN background shading by valence included.

### 4.3 COMPARISON WITH VALENCE-AROUSAL MAPS

On a qualitative level, we observe notable similarities between our internal latent space representations and valence-arousal organization from the psychology literature. We find that the learned emotion structure (Figure 2) across both LLMs appears to mimic the "V"-shaped parabolic emotion layout, with neutral emotions positioned approximately at the vertex and positive (and negative) valences clustering together along a dual "arm" structure. This semantic organization is visible in virtually all Gemma-2-9B layers and all but the early layers of Mistral-7B[2].

Beyond our qualitative support, we conduct a quantitative statistical test to compare our learned LLM representations against conventional valence-arousal maps. In particular, we compare the 2D MDS emotion embedding at each model layer with the map produced by third-party valence-arousal scores from the widely-used ANEW benchmark database from Bradley & Lang (1999). The ANEW coordinates (see Figure 2, right panel) for our statistical test come directly from human normative ratings, in which participants provided continuous valence and arousal scores for each lexical item. (We conduct our analyses using the 17 of the 28 classes in GoEmotions that are simultaneously covered in ANEW.) We then center both the MDS coordinates and the ANEW valence-arousal coordinates and align the MDS coordinates via scaled orthogonal Procrustes (single rotation and uniform scale), yielding fitted points corresponding to the MDS LLM latent representation. The test statistic is the Procrustes $R^2$, which measures the proportion of variance in the centered target configuration explained by the fitted configuration; significance is evaluated under a label-permutation null that shuffles the MDS emotion labels, recomputing the alignment and $R^2$ on each of 2,000 permutations. We compute one-sided $p$-values of the observed $R^2$ relative to this null under emotion label permutation. (This procedure is invariant to arbitrary MDS orientation/scale and does not assume metric validity of the separability matrix; inference derives entirely from permutation.)

We find that 36 of 42 layers in Gemma-2-9B and 17 of 32 layers in Mistral-7B (including all of the final 14 layers in Mistral-7B) exhibit statistically significant alignment with the ANEW valence-arousal scores, providing additional quantitative evidence that the learned LLM latent representations exhibit semantically coherent structure.

### 4.4 COMPARISON WITH NEURAL DATA

To further strengthen our claim of similar structure between human affective processing and LLM internal representations, we conducted an additional parallel analysis exploring whether similar se-

---

[2]These early Mistral-7B layers generally exhibit very low activation space emotion separability (see Apx. A), so the lack of semantic geometry here is consistent with the general observation that early Mistral-7B layers do not yet encode any coherent affective structure.

mantic representations of emotion emerge in human brainwave data. Using affective emotional data from 123 human subjects across 32 EEG channels (28 individual clips covering nine distinct emotions) from the FACED dataset (Chen et al., 2023), we investigate latent structure in the internal emotional landscape present in human neural data. We employed an experimental setting designed to resemble our LLM analysis: we trained pairwise logistic regressions on emotional clip classification and translated statistically significant test discriminatory accuracies into neural distances. (Statistical significance was computed via label permutation with a 0.05 p-value cutoff to account for the inherent noise present in neural data.) To address the high dimensionality of human brainwave data, we binned the neural signals into mean and variance summary statistics corresponding to conventional neural frequency bands (delta, theta, alpha, beta, and gamma) before performing pairwise logistic regressions. The resulting distances informed a downstream classical MDS analysis depicting the latent geometry of emotions in affective EEG data.

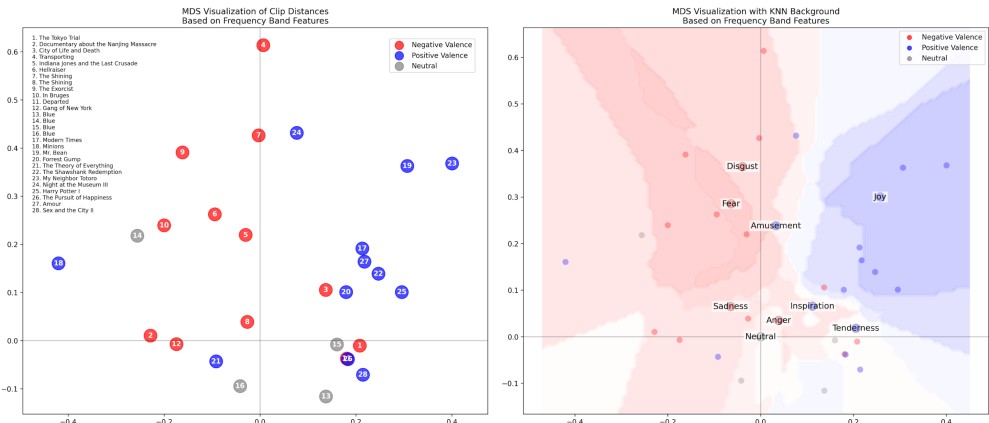

Figure 3: We find that a similar parabolic "V"-shaped emotion layout exists in human brainwave data, corroborating a link between human cognition and LLM affective emotion processing. Left panel depicts individual affective samples colored by valence; right panel depicts center of mass points for each emotion label alongside $k$NN shading by valence.

As shown in Figure 3, the embedding space exhibits a parabolic, "V"-shaped configuration. Neutral emotions are positioned near the vertex of the structure, while emotions with positive and negative valence again diverge along two distinct arms. This parallel suggests that the structural patterns observed in LLM representations may reflect principles of emotional representation that also characterize human neural activity, offering additional empirical support for the link between human affective cognition and learned representations in LLMs.

## 5 EVALUATING GEOMETRIC STRUCTURE

Our analyses in the previous section are based on a linear embedding assumption via classical MDS. To further probe the intrinsic geometry of LLM emotion representations, we conduct two quantitative analyses. First, we assess the dimensionality of pairwise emotion distances by examining the eigenspectrum of the classical MDS Gram matrix. Second, we investigate the manifold hypothesis using Isomap (Tenenbaum, 1997), constructing a $k$-nearest neighbor graph, estimating geodesic distances, and embedding them with classical MDS.

Eigenspectrum analyses (see Apx. A, Figure 7, left panel for an example), conducted across each individual layer of both Gemma-2-9B and Mistral-7B, reveal a generally diffuse rather than conclusively low-dimensional geometry. Under various monotone dissimilarity mappings ($D_{ij} = \mathrm{acc}_{ij}$ and $D_{ij} = \max(0, 2 \cdot \mathrm{acc}_{ij} - 1)$), the participation ratio remains high (Gemma-2-9B $\sim 17$, Mistral-7B $\sim 9$–$14$), indicating that variance appears spread across many eigenmodes[3]. This diffuse spectrum

---

[3]We note that as our dissimilarity metric is merely distance-like, we do observe occasional negative eigenvalues in our spectrum; however, only 1-2 negative eigendirections generally appear per layer, and these compose less than 1% of overall variance mass across virtually all layers exhibiting emotional separability.

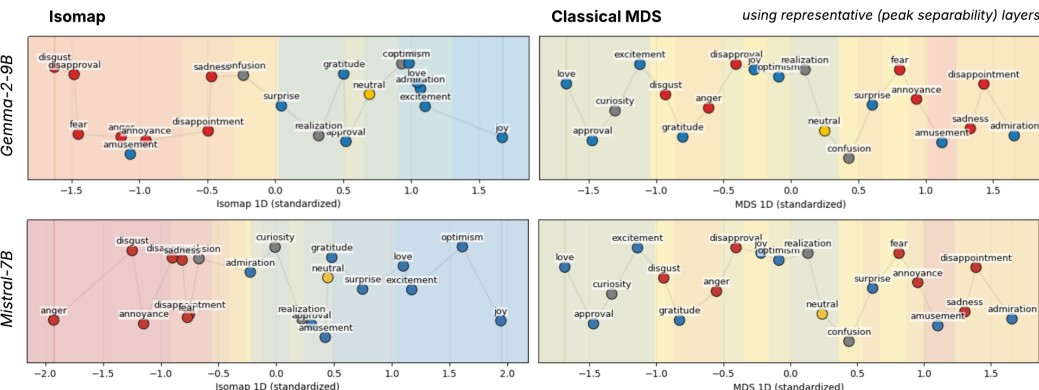

Figure 4: Isomap (*left*) appears to unroll parabolic structure in the emotion data manifold, resulting in improvements in capturing rank-1 structure over classical MDS (*right*). An artificial *y*-axis jitter is introduced for label visibility, and we include $k$NN background shading by valence to illustrate improved semantic coherence (*left*) under Isomap.

is consistent with what one would expect from a high-rank "bulk" component—akin to a Wigner-like distribution (Erdős et al., 2009)—from noise and task-irrelevant variation, with meaningful structure appearing only as deviations from this bulk. Thus, our goal is not to claim a globally low-dimensional manifold, but rather to identify statistically significant axes that separate from the bulk and correspond to affective structure. We do so via a statistical eigengap test where we evaluate scale-invariant ratios $r_k = \lambda_k / \lambda_{k+1}$ between ordered eigenvalues against a permutation-generated null. This null distribution is generated by shuffling the off-diagonal of $D$ (symmetry preserved), and we flag eigenvalues where the observed ratio falls in the high tail ($p_{\text{hi}} < 0.05$). In general, our permutation test results (assessing prominence from the eigenspectrum bulk) appear generally consistent with a diffuse overall structure and no definitive fixed rank. We do find that Gemma-2-9B shows a recurring significant first eigengap with 16 out of 42 layers meeting a significance threshold at $k = 1$, consistent with a dominant valence-like axis; we note, however, that this gap does not by itself establish any sort of fixed intrinsic dimension, but rather points to a recurring tendency toward a prominent individual axis of variation.

Our complementary Isomap evaluations allow us to probe for potential nonlinear manifold structure. For each layer, we constructed $k$NN[4] graphs over the pairwise separability matrix $|d'|$, computed geodesic distances, and compared Isomap against a classical MDS (Euclidean) baseline via two diagnostics: (i) relative trustworthiness (see Apx. B for formal definition) of local neighborhoods, and (ii) divergence between geodesic and Euclidean distances. On the overall trustworthiness front, we find generally minimal (i.e., $< 0.03$) differences in trustworthiness between Isomap and classical MDS across ranks and choices of k–with the exception of the rank-1 embedding, for which Isomap demonstrates a median trustworthiness increase of 0.161 in Gemma-2-9B and 0.127 in Mistral-7B. We speculate that this rank-1 improvement is due to the aforementioned parabolic "V"-shaped structure of the emotion data manifold, which Isomap naturally captures in "unrolled" form in the 1D setting as shown in Figure 4.

In terms of discrepancies between geodesic (Isomap) vs. Euclidean (classical MDS) distances, we observe evidence consistent with modest nonlinearity. Specifically, distance ratios (i.e., geodesic / Euclidean) span from 1.00 to 1.80 from the tenth to ninetieth percentiles in Gemma-2-9B and 0.80 to 1.42 in Mistral-7B[5]. These results suggest that the LLM emotion spaces are almost Euclidean: high-rank, diffuse, and lacking strong low-dimensional curvature. The manifold nonlinearities that do appear in higher rank spaces appear consistent with the natural parabolic bending of valence–arousal space (e.g., see Apx. A, Figure 7, right panel, in addition to Figure 4); the emotion data manifold appears to exhibit a natural mode of curvature (due to the correlation between valence intensity and arousal) that manifests in LLM latent representations. On the whole, however, nonlinear manifold

---

[4]We select the Isomap $k$ parameter to maximize trustworthiness.

[5]We select the embedding dimension $d$ via an elbow-esque method where we identify the first residual variance drop less than 0.02.

structure does not appear strong enough to render Euclidean representations ineffective as a basis for analysis.

Table 1: Mean trustworthiness improvement using Isomap over classical MDS per rank $d$

| Model | $d = 1$ | $d = 2$ | $d = 3$ | $d = 4$ | $d = 5$ | $d = 6$ | $d = 7$ | $d = 8$ |
|---|---|---|---|---|---|---|---|---|
| Gemma | 0.1552 | -0.0013 | -0.0010 | 0.0106 | 0.0127 | 0.0199 | 0.0238 | 0.0223 |
| Mistral | 0.0994 | -0.0241 | -0.0263 | -0.0113 | -0.0046 | 0.0005 | -0.0036 | -0.0070 |

## 6 APPLICATIONS IN UNCERTAINTY QUANTIFICATION

During our pairwise logistic regression tests, we also saved activations corresponding to LLM misclassifications. Unlike the correct activation setting, where each activation is associated with a single emotion, each misclassification is linked to two emotions: the model outputted emotion, and the ground-truth GoEmotions label. Under a naive assumption of no semantic affective linkages, one might expect the misclassified activations to lie close in activation space to the "correct" activations for the LLM-outputted class. However, we instead observe a remarkable phenomenon where these misclassified activations instead lie in between the two emotions to which they are linked[6]; that is, the "misclassified" activations hover much closer to the separating hyperplane between the two emotions (as defined by the corresponding pairwise logistic regression) than the "correct" activations do. We note that this phenomenon is not tautological, as we only evaluate distances to the separating hyperplane in terms of correct *test* samples and misclassified activations, neither of which were included in the original logistic regression training data.

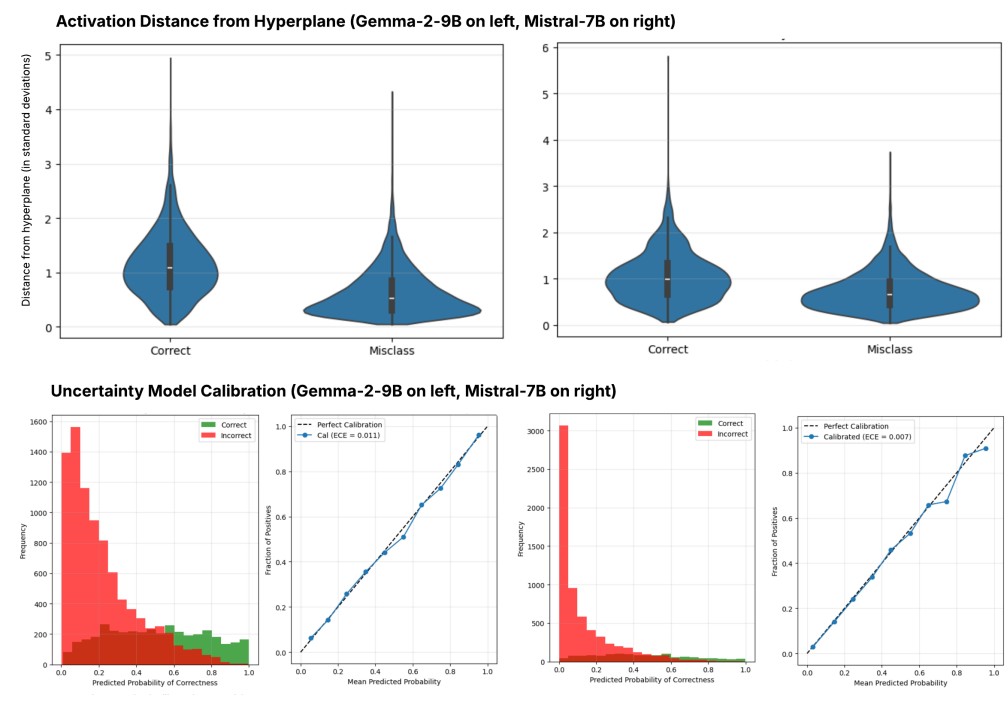

Figure 5: Upper panels depict activation distance from separating hyperplane, with misclassifications lying closer to the boundary than correct classifications. Lower panels depict test results of calibrated uncertainty quantification models, demonstrating robust calibration quality.

We can exploit this observation to generate practical utility by training a second round of pairwise logistic regression models on per-layer activation-to-hyperplane distances. Our work in this predictive *uncertainty quantification* setting is motivated by the potential benefit of detecting LLM

---

[6]Early layers lie slightly closer to the original prompt emotion, and late layers lie slightly closer to the erroneous outputted emotion (see Apx. A).

emotion processing misclassifications–the core concept, here, is to use the distance in activation space associated with an LLM prediction from the separating hyperplane between two classes as a measure of LLM "confidence" in that prediction. We restrict our analyses to emotion pairs with at least 25 correct and 25 misclassified samples in the data, resulting in 180 viable binary classification problems (3,858 correct samples vs. 8,748 misclassified) for Gemma-2-9B and 122 for Mistral-7B (5,254 correct vs. 15,216 misclassified) under a 60:20:20 train-val-test split. We use validation data in order to calibrate the logistic regression models, mapping raw hyperplane distances across all layers to well-calibrated probability estimates of correctness.

Our trained uncertainty models post 77.6% accuracy (0.813 AUC-ROC) on Gemma-2-9B and 85.7% accuracy (0.871 AUC-ROC) on Mistral-7B, compared to majority-class baselines of 69.4% and 82.2%, respectively. Importantly, post-calibration quality in terms of predicted correctness probability is strong: we find expected calibration errors of 0.011 (Gemma-2-9B) and 0.007 (Mistral-7B) on held-out test data. These results show that geometry-informed separating hyperplane distance–based regressions yield well-calibrated, discriminative uncertainty estimates in an LLM classification setting.

## 7 GENERALIZATION: LLAMA-3-70B-INSTRUCT

To further assess the generalizability of our findings, we conducted a complete replication of our pipeline on LLaMA-3-70B-Instruct (AI@Meta, 2024), a substantially larger (70B parameter) and instruction-tuned model. Instruction tuning is an important axis of variation: unlike the base pre-training objective, instruction tuning introduces supervised alignment and human-feedback–driven optimization, which are known to reshape representational structure in ways that may alter semantic geometry. Demonstrating stability of affective geometry under such changes therefore provides a more rigorous test of generalization.

LLaMA-3-70B-Instruct achieves a zero-shot emotion classification accuracy of 21.3% on GoEmotions, outperforming both Mistral-7B (13.7%) and Gemma-2-9B (19.4%). Classical MDS embeddings reveal the same macroscopic affective layout observed in the earlier models: positive and negative emotions diverge along two "arms," with neutral emotions located near the geometric vertex. Figure 6 depicts results from all core LLaMA analyses (separability, MDS embeddings, Isomap embeddings, and ANEW alignment). We conducted the same scaled orthogonal Procrustes test using the ANEW valence–arousal coordinates. LLaMA-3-70B-Instruct shows alignment across a wide band of layers: 34 of 80 layers exhibit significant 2D Procrustes alignment with $p_{2D} < 0.05$, primarily concentrated among later layers with well-formed emotion separability. Moreover, as with the smaller models, Isomap recovers the expected parabolic structure more explicitly in low-rank embeddings. Trustworthiness scores for LLaMA-3-70B-Instruct improve in the rank-1 setting relative to classical MDS, consistent with the "unrolling" of a curved valence–arousal manifold (with higher-rank embeddings showing minimal differences, again consistent with previous results). In addition, we replicated the uncertainty quantification pipeline described in Section 6, training logistic regressions on LLaMA's distances-to-hyperplane across all layers. Despite LLaMA-3-70B-Instruct's distinct training regime and large parameter count, the method again produces strong discriminative and calibration performance. Notably, the misclassified samples again lie closer to pairwise separating hyperplanes than correct samples, mirroring the geometric confidence patterns seen in Gemma and Mistral. On held-out test data, our trained uncertainty model on LLaMA posts 80.1% accuracy (0.822 AUC-ROC) with expected calibration error of 0.011; for comparison, baseline accuracy is 75.6%. These results help reinforce the generality of experiments across architectures, scales, and training paradigms; further details are included in Apx. A, Figure 14.

## 8 DISCUSSION

Our analyses reveal that LLM emotion representations exhibit coherent geometric structure that aligns with affective models from psychology. We further present initial evidence for similar latent structure in human brainwave data. Moreover, we show direct utility in the form of calibrated uncertainty models for LLM emotion processing, which leverage representation geometry to provide reliable estimates of predictive confidence.

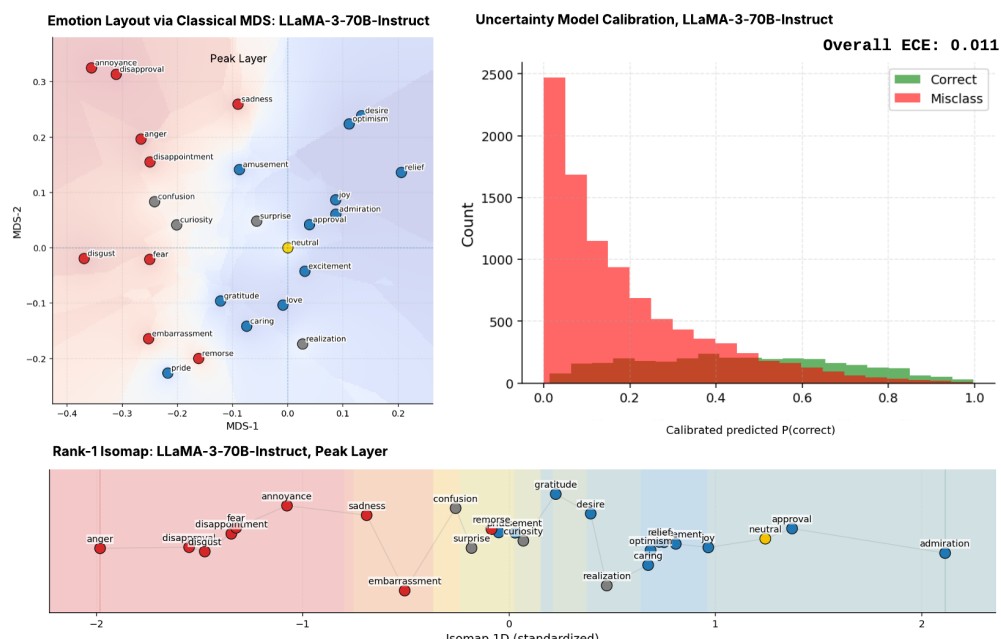

Figure 6: Corroboration of key results on LLaMA-3-70B-Instruct. Top-left: MDS emotion layout. Top-right: uncertainty model calibration. Bottom: Isomap embedding reflecting unrolled curvature.

**Limitations**   First, it remains unclear whether the observed patterns may generalize beyond the tested LLMs. Expanding our analysis to a wider range of larger models and a more diverse set of language model families is an important direction for future work. Another avenue for future investigation is a systematic study of the effects of emotion-focused fine-tuning. Second, our use of pairwise logistic regressions for tractability purposes (i.e., to avoid grappling with heavily class-imbalanced high-dimensional activation covariances) in conjunction with MDS and Isomap may not fully capture potential geometric intricacies; distributional analyses as well as advanced geometric and topological data analysis tools could possibly reveal nuanced relationships in future work. Third, our primary GoEmotions dataset could have cultural or linguistic biases as well as a narrower set of writing modes that may constrain the generality of our findings. Extending the analysis to more naturalistic prompts and to sub-layer components (e.g., attention vs. MLP) would also be valuable future work, but is outside the scope of the present study.

Our secondary analysis regarding similar patterns in human brainwave data also exhibits several limitations that we hope to address in future work. While our results present first evidence for the alignment of latent structure in LLM representations and human brainwave data, this does not constitute evidence of shared cognitive encoding. Furthermore, our observations are based on a single dataset; a study with larger scope is an interesting avenue for future work. An analysis of the impact of cultural biases in the dataset was likewise beyond the scope of the present paper.

**Broader Impacts**   The parallels we uncover between LLM and human affective representations highlight potential convergences between artificial and natural cognition, suggesting that geometric priors rooted in human psychology may inform future model interpretability frameworks. On the application side, our uncertainty quantification method illustrates a principled way to identify when LLMs may be likely to misclassify, with potential downstream relevance to selective prediction, human-in-the-loop systems, hallucination detection, steering model emotional tone and other safety-critical applications.

**Reproducibility Statement**   To facilitate replication of our analyses, we provide an anonymized repository containing code to reproduce all experiments, figures, and statistical tests at this link. Detailed descriptions of experiments, preprocessing steps, and statistical analyses are included throughout the manuscript and in Appendix B.

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

## A ADDITIONAL EXPERIMENTAL RESULTS

Figure 7 depicts further results from our analyses in Section 5. Figures 8, 9, and 10 depict further results from our pairwise logistic regression and classical MDS experiments. The mean emotional separability between discrete emotion classes for each layer in Figure 8 is measured via pairwise logistic regression test accuracies.

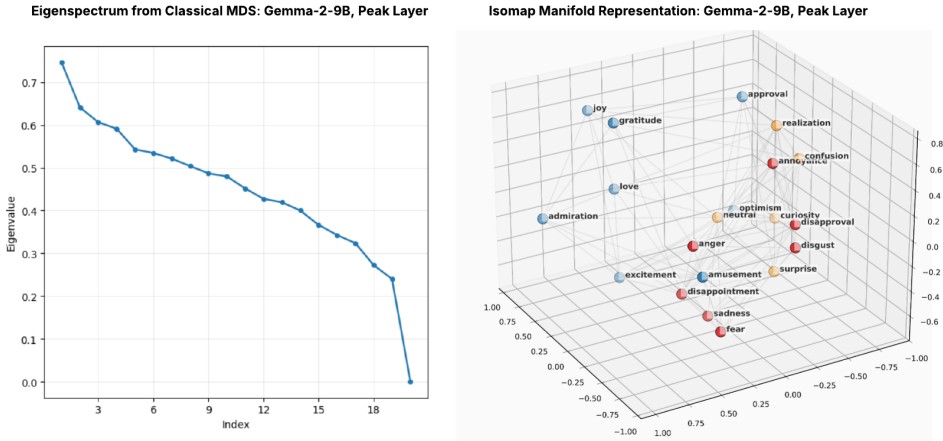

Figure 7: Representative examples of a classical MDS eigenspectrum (*left*) and an Isomap embedding visualization (*right*).

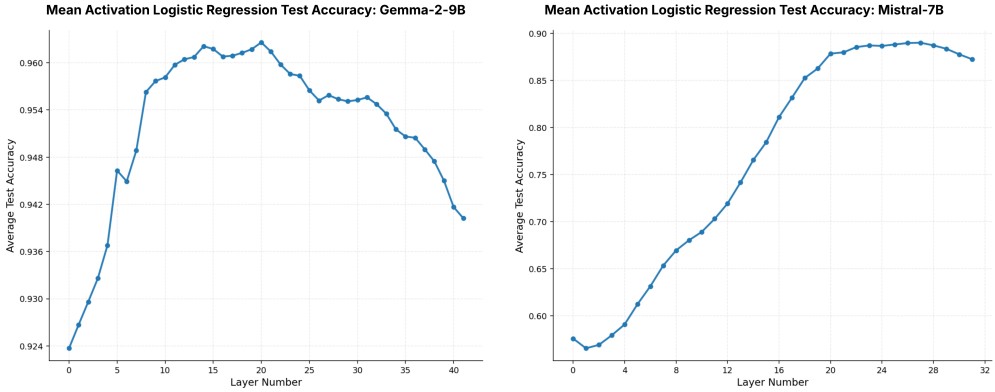

Figure 8: Mean emotional separability across layers for Gemma-2-9B (left) and Mistral-7B (right). Separability increases with depth before tapering off in the final layers, with Gemma-2-9B exhibiting higher separability than Mistral-7B.

Figure 10 depicts results from our statistical alignment test between the classical MDS embeddings and established valence-arousal scores, showing results ($R^2$ and $p$-values) for two statistically significant layers in Gemma-2-9B and Mistral-7B. Figure 11 depicts layer-by-layer trends from our pairwise logistic regressions on the out-of-sample misclassified pairs. More specifically, lower (or more negative) y-axis values signify that activations are closer to the ground truth input emotion, while higher values signify that activations are closer to the output emotion. Misclassified activations, as discussed in Section 6, lie closer to the separating hyperplane than correct activations; as shown in Figure 11, activations trend away from the ground truth input emotion and toward the model output emotion as model layers progress. Figure 14 depicts additional results from our experiment generalizing our findings to LLaMA-3-70B-Instruct.

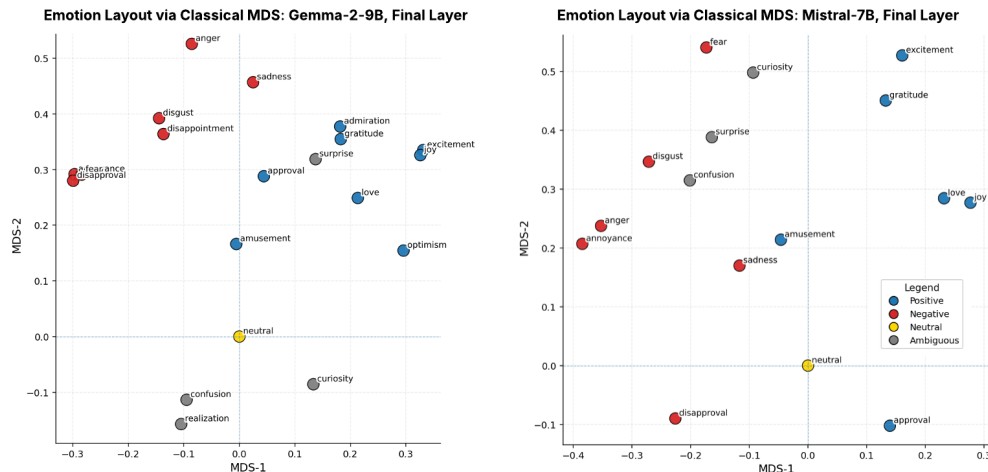

Figure 9: MDS emotion layouts for final layers in Gemma-2-9B (*left*) and Mistral-7B (*right*).

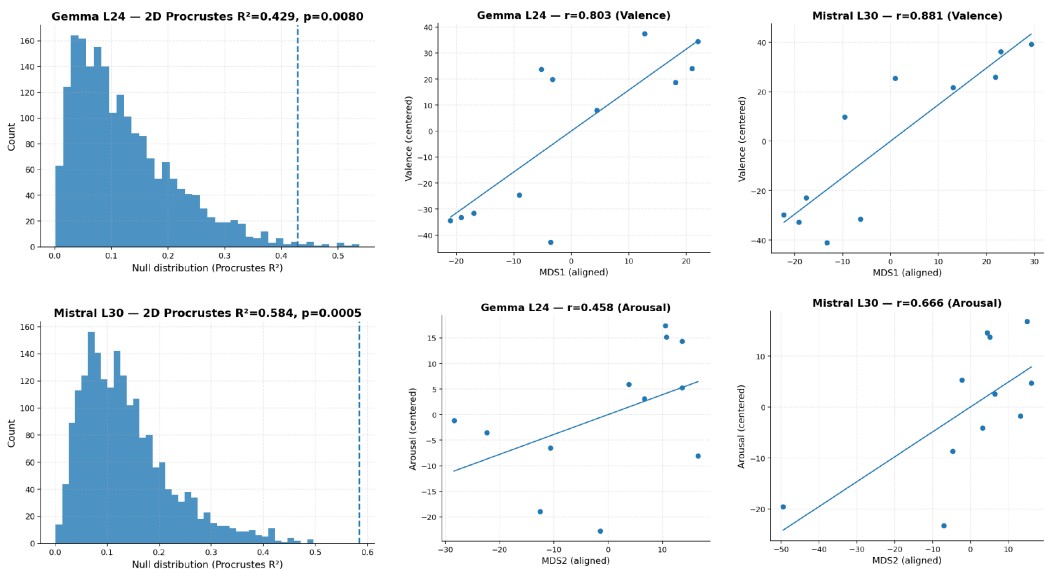

Figure 10: $R^2$ and $p$-values for the classical MDS LLM latent representations vs. established valence-arousal scores.

### UMAP VISUALIZATIONS OF DISTANCE GEOMETRY

To complement our classical MDS and Isomap analyses, we additionally conducted a supplementary analysis using UMAP (McInnes et al., 2018). For each model (Gemma-2-9B, Mistral-7B, and LLaMA-3-70B-Instruct) and each layer, we performed a small UMAP hyperparameter sweep over the number of neighbors $n_{\text{neighbors}} \in \{3, \ldots, 12\}$ as opposed to proceeding directly to classical MDS. For each setting we computed the trustworthiness (Venna & Kaski, 2001) of the resulting embedding with respect to the original distance matrix, and selected the $n_{\text{neighbors}}$ that maximized trustworthiness. Visualizations of the 2D UMAP case for representative layers are shown in Figure 12. We also verified the presence of statistically significant alignment patterns between these UMAP embeddings and the ANEW reference scores, with 34 significant layers in LLaMA-3-70B-Instruct, 10 in Mistral, and 41 in Gemma ($p < 0.05$).

Across these settings, UMAP recovers general semantic clustering classical MDS, with positive, neutral, and negative sentiments exhibiting relatively coherent and clustered layouts. These results

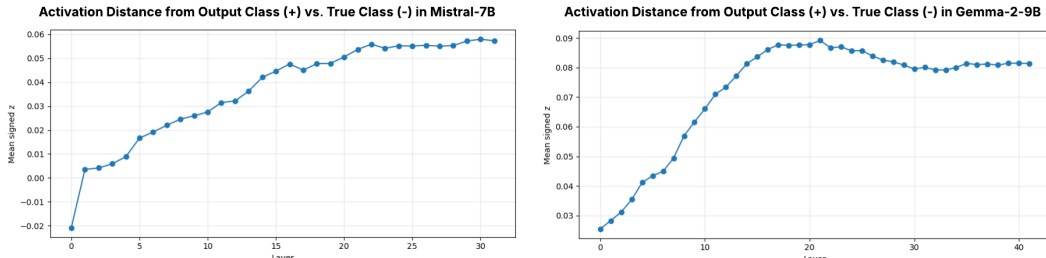

Figure 11: Layer-by-layer activation distances from output class vs. ground-truth input class. Higher values signify relatively closer distances to the model output emotion class.

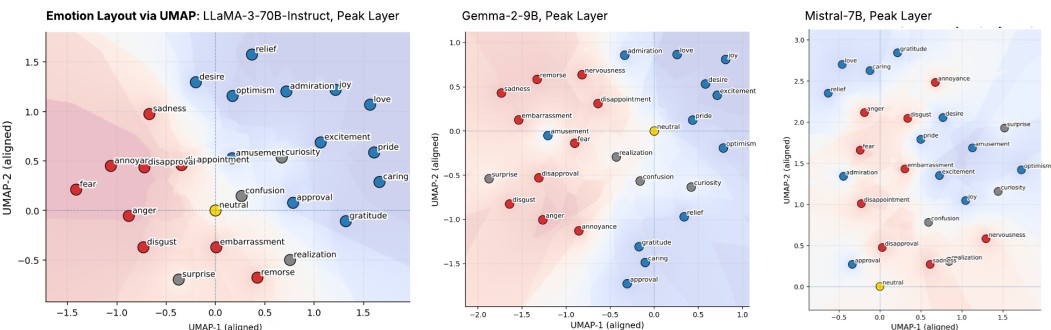

Figure 12: UMAP visualizations for Gemma-2-9B, Mistral-7B, and LLaMA-3-70B-Instruct.

indicate that our observed semantic structure is stable under a nonlinear, neighborhood-preserving embedding applied directly to the pairwise distance geometry.

### COSINE-DISTANCE EXPERIMENT

Our main analyses define dissimilarities via pairwise logistic-regression separability, which couples the inferred geometry to a particular supervised probing pipeline. To test whether our conclusions depend critically on this choice, we performed an additional ablation based on a purely geometric cosine-distance metric over the mean-pooled activations.

Concretely, for emotion class across layers, we first computed the mean activation vector by averaging the mean-pooled hidden states over all correctly classified examples of that emotion (as in the main pipeline). We then constructed an alternative distance matrix

$$D_{ij}^{\cos} = 1 - \cos\big(\bar{h}_i, \bar{h}_j\big),$$

where $\bar{h}_i$ and $\bar{h}_j$ denote the mean activation vectors for emotions $i$ and $j$, and $\cos(\cdot, \cdot)$ is the cosine similarity. As before, diagonals were set to zero, missing entries were imputed by the global mean, and the matrix was symmetrized.

Using $D^{\cos}$, we then repeated the classical MDS procedure; Figure 13 shows an example of these cosine-based embeddings. Qualitatively, we again observe clear semantic clustering; quantitatively, recomputing our Procrustes $R^2$ ANEW alignment test, we also observe broad statistically significant alignment

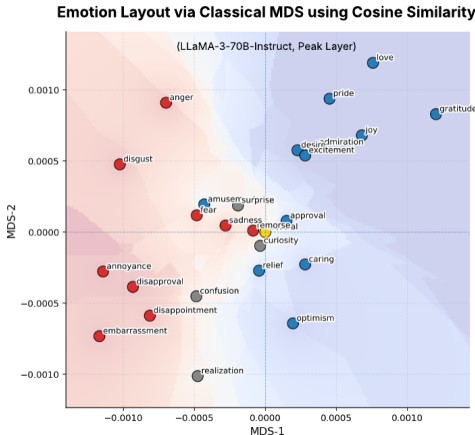

Figure 13: Classical MDS embeddings obtained from cosine-based dissimilarities between mean activation vectors.

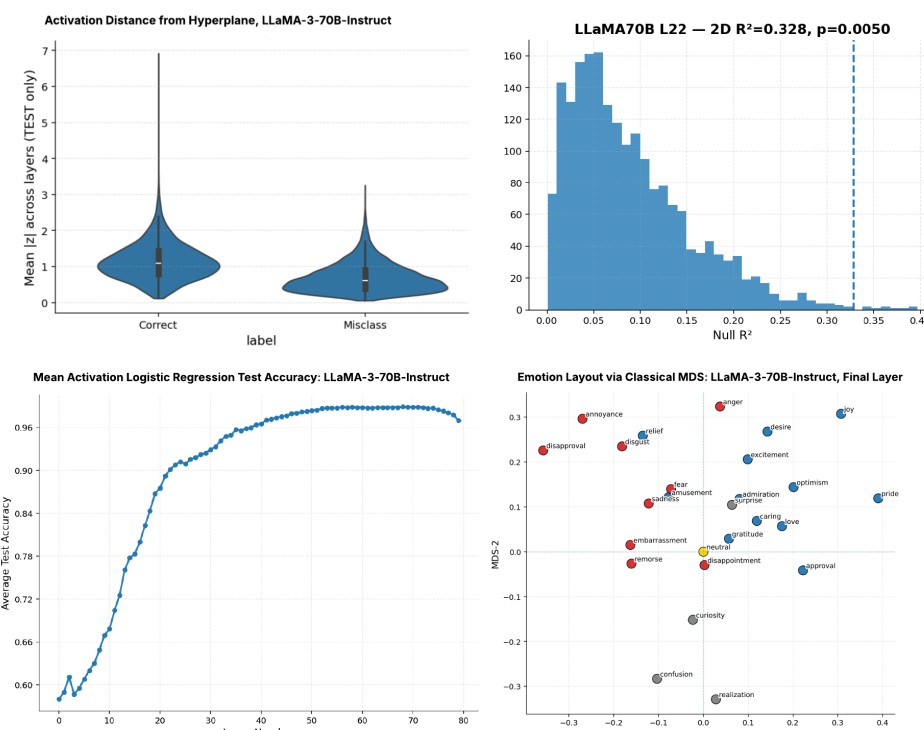

Figure 14: Additional results from experiments on LLaMA-3-70B-Instruct. Top-left: activation distance from hyperplane for correct vs. misclassified samples. Top-right: statistical significance result for ANEW alignment. Bottom-left: mean logistic regression accuracy by layer. Bottom-right: classical MDS emotion layout for final layer.

across all layers. Taken together, this cosine-distance experiment helps support the robustness of our main findings by corroborating our affective representation geometry.

## B   Additional Experimental Details

**GoEmotions Data**   The GoEmotions corpus used in this study contains the following emotion categories: admiration, amusement, anger, annoyance, approval, caring, confusion, curiosity, desire, disappointment, disapproval, disgust, embarrassment, excitement, fear, gratitude, grief, joy, love, nervousness, optimism, pride, realization, relief, remorse, sadness, surprise, and neutral.

For Mistral, the correct sample data distribution included 47 samples of admiration, 1293 of amusement, 509 of anger, 228 of annoyance, 275 of approval, 31 of caring, 123 of confusion, 312 of curiosity, 29 of desire, 93 of disappointment, 373 of disapproval, 213 of disgust, 86 of embarrassment, 218 of excitement, 201 of fear, 670 of gratitude, 203 of joy, 1048 of love, 35 of nervousness, 60 of optimism, 36 of pride, 89 of realization, 40 of relief, 744 of sadness, 291 of surprise, and 103 of neutral.

For Gemma, the correct sample counts were 742 for admiration, 1211 for amusement, 455 for anger, 411 for annoyance, 216 for approval, 95 for caring, 481 for confusion, 675 for curiosity, 88 for desire, 439 for disappointment, 1148 for disapproval, 177 for disgust, 60 for embarrassment, 304 for excitement, 154 for fear, 678 for gratitude, 284 for joy, 300 for love, 37 for nervousness, 208 for optimism, 36 for pride, 101 for realization, 55 for relief, 58 for remorse, 235 for sadness, 318 for surprise, and 2212 for neutral.

For LLaMA-3-70B-Instruct, the correct-sample distribution included 1105 samples of admiration, 1555 of amusement, 472 of anger, 222 of annoyance, 518 of approval, 246 of caring, 407 of confusion, 689 of curiosity, 67 of desire, 265 of disappointment, 1225 of disapproval, 240 of disgust, 70 of embarrassment, 249 of excitement, 182 of fear, 1135 of gratitude, 223 of joy, 341 of love, 203 of

optimism, 47 of pride, 84 of realization, 85 of relief, 103 of remorse, 333 of sadness, 307 of surprise, and 1233 of neutral.

As mentioned in Section 4, we also conducted exploratory experiments with Qwen and LLaMA models. Specifically, we ran our affective classification workflow on Qwen2.5-7B and LLaMA-2-7B, but found that neither model produced sufficient correct classifications (i.e., $>100$ across more than three individual categories) on the GoEmotions dataset which are requisite for our downstream analyses.

**Classification Routine**   We analyze Gemma-2-9B, Mistral-7B, and LLaMA-3-70B-Instruct using AutoModelForCausalLM at float16 precision. For single-label predictions, each classification prompt shuffles the vocabulary of emotions to reduce positional bias. The prompt template is: "Classify this text into exactly one emotion from this list: ... Text: {text} Emotion:". The first token generated after the word "Emotion:" is decoded and matched to the enumerated set, with unmatched predictions skipped from activation saving. To capture internal activations, we register forward hooks on every transformer block. The hooks extract hidden states, apply mean pooling across the sequence dimension to produce a single vector per layer, and store outputs as detached tensors. The shuffled emotion order is reused to maintain consistency between prediction and activation passes.

Balanced pairwise training is carried out by constructing equal-sized datasets for every emotion pair $(e_i, e_j)$. Concatenated data matrices $\mathbf{X} \in \mathbb{R}^{2m \times H}$ with binary labels $\mathbf{y} \in \{0, 1\}$ are split in an 80/20 ratio with stratification. Logistic regression models with maximum iterations of 1000 are trained and evaluated, and we record training and test accuracy, decision margins, per-point correctness, and sample counts to assist in downstream analyses.

**Statistical Alignment Test**   As discussed in 4.3, to align model-derived spaces with our external valence-arousal ratings for our statistical alignment test, we use an orthogonal Procrustes transformation. Let $Y \in \mathbb{R}^{n \times 2}$ denote the valence-arousal matrix, centered column-wise, and let $X$ denote the corresponding model embedding. The alignment is defined as

$$\min_{R \in \mathbb{R}^{2 \times 2}, R^\top R = I, \, a \in \mathbb{R}} \|a X_c R - Y_c\|_F^2,$$

where $X_c$ and $Y_c$ are row-centered. The solution is obtained by singular value decomposition $X_c^\top Y_c = U \Sigma V^\top$, with $R = U V^\top$ and $a = \mathrm{tr}(\Sigma)/\|X_c\|_F^2$. The alignment coefficient of determination is reported as

$$R_{\mathrm{proc}}^2 = 1 - \frac{\|a X_c R - Y_c\|_F^2}{\|Y_c\|_F^2}.$$

Significance of alignment is assessed via permutation tests. Row labels of $Y$ are permuted and $R_{\mathrm{proc}}^2$ is recomputed for $T = 2000$ shuffles. With observed value $R_{\mathrm{obs}}^2$, the $p$-value is

$$p = \frac{1 + \sum_{t=1}^{T} \mathbf{1}\{R_{(t)}^2 \geq R_{\mathrm{obs}}^2\}}{T + 1},$$

with reproducibility guaranteed via fixed random seed. After alignment, axis-wise correlations are calculated between aligned $X_{\mathrm{hat}}$ and $Y_c$, with Pearson correlations $r_{\mathrm{val}}$ for valence and $r_{\mathrm{aro}}$ for arousal; these metrics are shown in Figure 10. These correlations are also subjected to permutation tests that re-align for each shuffle, ensuring axis assignment does not bias significance.

Practical safeguards include mean pooling across sequence length to prevent padding sensitivity, imputation of missing similarity values by the global mean to avoid distortions, clamping of negative eigenvalues from double-centering to zero for stability, and centering of both $X$ and $Y$ prior to Procrustes analysis so that explained variance is measured relative to mean-centered data. Orthogonal Procrustes rotations may arbitrarily swap or rotate axes, so axis correlations are always computed post-alignment with refitting under permutation to ensure robustness.

For additional context, we also include a depiction of the result of applying rank-1 Isomap directly on the rank-2 ANEW valence-arousal scores. As shown in Figure 15, we find that Isomap produces a smooth one-dimensional semantic progression, mirroring the effect we observe in our LLM embeddings.

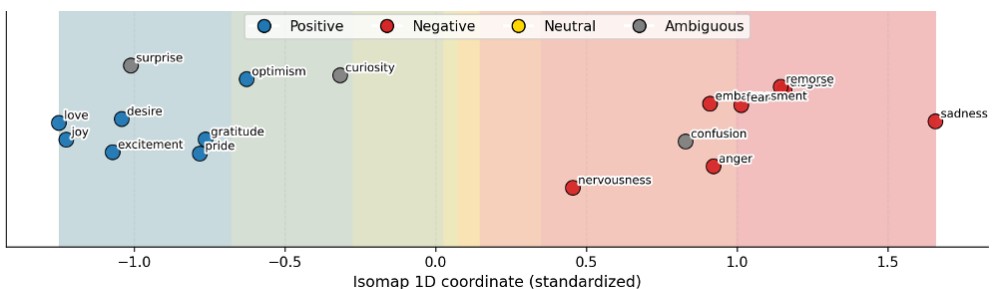

Figure 15: Isomap embedding of the ANEW valence–arousal space.

**Trustworthiness Definition** Trustworthiness (Venna & Kaski, 2001) measures how well local neighborhoods in the high-dimensional data are preserved in a low-dimensional embedding. It ranges from 0 to 1, with 1 indicating perfect preservation of $k$-nearest neighbors. More specifically, we compute:

$$T(k) \; = \; 1 - \frac{2}{n\,k\,(2n - 3k - 1)} \sum_{i=1}^{n} \sum_{j \in U_i} \big(r_i(j) - k\big), \tag{1}$$

where $n$ is the number of samples, $U_i = N_k^{\text{low}}(i) \setminus N_k^{\text{high}}(i)$ are intrusions (neighbors in the embedding but not in the original space), and $r_i(j)$ is the rank of $j$ with respect to $i$ in the original space.

**More on Uncertainty Quantification** To make our uncertainty quantification methods more concrete, we include a more detailed description of their use case. For a given labeled GoEmotions prompt, we consider the likelihood that the LLM output is indeed the ground truth against an alternative where the LLM output is instead another erroneous emotion. Our calibrated logistic regression model (trained across hyperplane distances from all activation layers) outputs an estimated probability that the LLM output matches the ground truth emotion.

Our results (as discussed in Section 6) indicate that the probability produced is not only discriminative of correct versus incorrect classifications, but also well-aligned with empirical frequencies of correctness. In other words, a sample assigned a predicted probability of 0.8 is correct roughly 80% of the time. This calibration property is quantitatively reflected in the low expected calibration error (ECE) achieved by our models: 0.011 for Gemma-2-9B and 0.007 for Mistral-7B on held-out test data. These low ECE values confirm that predicted probabilities can be interpreted directly as trustworthy uncertainty estimates.

**Compute** All experiments were conducted using NVIDIA H200 GPUs with PyTorch 2.5.1 and CUDA 12.1.

**LLM Statement** Large language models were used to assist in the preparation of this work. Specifically, they were employed for code generation (via Cursor) and for light formatting of draft text. All experimental design decisions, analyses, and interpretations were made by the authors.

**Licenses**

| Model/Dataset | License | Notes |
|---|---|---|
| Mistral-7B-v0.1 (Jiang et al., 2023) | Apache License 2.0 | See here for license |
| Gemma-2-9B (Team et al., 2024) | Gemma License | See here for license |
| LLaMA-3-70B-Instruct (AI@Meta, 2024) | Llama 3 Community License | See here for license |
| GoEmotions (Demszky et al., 2020) | CC BY 4.0 (data) | See here for license |
| FACED (EEG) (Chen et al., 2023) | CC BY 4.0 (data) | See here for license |

Table 2: Licenses for models and datasets used in our experiments.

