# OpenReview forum: "Latent Structure of Affective Representations in Large Language Models"
_ICLR.cc/2026/Conference — Submitted to ICLR 2026_

### Official Review · Reviewer_c3Kz · 2025-10-29

**Soundness:** 4
**Presentation:** 4
**Contribution:** 4
**Rating:** 8
**Confidence:** 5

**Summary:**

This paper explores the geometric structures of latent representations of emotions in large language models, and provides evidence that they show some alignment with valence-arousal models of emotions from psychology and geometric structures observed in human brainwave data.

They employ the GoEmotions dataset of  English-language Reddit comments which have been manually annotated with one of 27 fine-grained emotion categories (or labelled as neutral). These sentences are passed though the Gemma-2-9B and Mistral-7B transformer language models and mean-pooled activations at each layer were recorded. They perform pairwise classification analysis to test the linear separability of emotion representations, finding high test accuracy (>0.9) in Gemma-2-9B across al layers and moderate test accuracy (0.55-0.89) in Mistral-7B.

In addition, they perform dimension reduction using C-MDS and ISOMAP and visualse these activation, observing weak geometric alignment with valence-arousal coordinates (from the ANEW dataset). They also provide numerical quantification of this using the $R^2$ statistic of a scaled orthogonal Procrustes alignment.

They perform similar analysis of neural EEG data and find similar conclusions, and demonstrate how the geometry of misclassified emotion can be leveraged for uncertainty quantification.

**Strengths:**

This paper has a well-stated goal, a clear and effective methodology, and is very clearly presented and written. The paper provides additional insight into how latent concepts are represented in the internal activations of transformer models, and is a valuable addition to this line of interpretability research

In particular, the paper is well-scoped and sets out a clear hypothesis: "Do LLMs develop coherent internal representations of emotions that align with the valence-arousal model?". The methodology is unambiguous and clearly explained, and the evidence that is provided for the hypothesis is presented in an objective manner that tells a clear story and is not overstated. The graphical displays of the emotion representations are very insightful and nicely presented and quantitive analysis backs up the conclusions which are suggested by the plots.

The analysis of the neural EEG data is also very enlightening, and may suggest additional hypotheses that might be investigated by future researchers. The investigation of the the inherent ambient and intrinsic dimension of the representations, and into the non-linearity of the representations also helped complete by understanding of the authors investigations.

Overall, I really enjoyed this paper and would recommend it for acceptance.

**Weaknesses:**

I think the authors did a good job of defining their scope and thoroughly investigating their hypothesis within this scope. For this reason, I have no weaknesses to mention.

**Questions:**

I would be interested to know whether you manually inspected your nearest-neighbour graphs when performing ISOMAP to check for short-circuits? I also think it would be interesting to show the ANEW valence-arousal plots alongside the C-MDS plots, and and the result of applying ISOMAP to it alongside the ISOMAP plots.

---

> ### Author Response · Authors · 2025-11-22
> **Response to reviewer c3Kz**
>
> We thank the reviewer for the careful read of our paper and the positive feedback. We believe that our paper has strengthened even further over the course of the review process.
>
> > I would be interested to know whether you manually inspected your nearest-neighbour graphs when performing ISOMAP to check for short-circuits? I also think it would be interesting to show the ANEW valence-arousal plots alongside the C-MDS plots, and and the result of applying ISOMAP to it alongside the ISOMAP plots.
>
> Yes. In addition to using trustworthiness as our formal quantitative diagnostic, we also manually inspected the k-NN graphs for representative layers across all three models (Gemma-2-9B, Mistral-7B, and the newly added LLaMA-3-70B-Instruct analysis). We also now include the ANEW valence-arousal plot alongside the C-MDS plots, as well as the result of applying ISOMAP to the ANEW valence-arousal embeddings.

---

> > ### Comment · Reviewer_c3Kz · 2025-11-24
> >
> > Thank you for your response. I continue to strongly endorse this paper.

---

### Official Review · Reviewer_kBVa · 2025-10-31

**Soundness:** 1
**Presentation:** 2
**Contribution:** 3
**Rating:** 2
**Confidence:** 4

**Summary:**

This paper seeks to understand the geometry of emotion representation in LLMs. It does this by using the accuracy score of the pair-logistic classifier over the hidden states of the model when prompted to classify the emotions of a text. It then uses MDS and other latent space techniques on the distance matrix constructed from the pairwise accuracies to get the embeddings space. This method is repeated on EEG data and the resulting 2D map of emotion is compared between the two types of data. Finally a emotion classification technique is proposed that incorporates some of the techniques from earlier in the paper.

**Strengths:**

The paper is very ambitious in its scope. It tries to find a latent geometry of emotions in LLMs and how it maps to EEG data. Both of these are important and interesting problems that the community needs to tackle. Understanding how LLMs process emotions is important to understanding how they interact with the human-side of language. The method in section 6 is an interesting approach to improve LLMs ability to do emotion classification. The paper acknowledges its limitations well (though despite this, some of them are core weaknesses nonetheless).

**Weaknesses:**

The paper is very ambitious overall. However, I fear that the paper tends to overclaim results instead of properly explaining what their results are showing.
-	By asking the LLM to classify the emotion of the text, there is the worry that you are “priming” the network to focus on emotion and thus are not looking at how it processes emotions naturally when no attention is focused on that element of text.
-	The paper examines hidden states but not effects of each sub-layer
-	Results are only on two LLMs
-	GoEmotions is on reddit dataset which has one mode of writing. Would prefer to see the results of this analysis on multiple emotion classification datasets to see if the results generalize across different modes of writing.
-	Figure 2 appears to show the geometry of the MDS accuracy-space, not the intrinsic emotional geometry of the LLM’s hidden states. This visualization therefore reflects how easily separable emotions are under linear probes, rather than how emotions are internally organized in the model. Moreover, representing each emotion as a single point, rather than distributions or clouds of examples, obscures whether this organization is consistent within and across emotion labels.
-	It is not clear how VAD scores were produced for the GoEmotions dataset.
-	The comparison with EEG data is again a comparison logistic regression accuracy scores (how separable are the emotions in the EEG data). I am not sure this shows any mapping between how the two types of data represent emotions. At most it seems to show that these emotions are similarly linearly separable. That is an interesting result, but not the same as what is being claimed in the paper.
-	Similarly, from my understanding of reading Section 5, the geometric structure analyzed is of the separability matrix, not of the hidden-states themselves.
-	Section 6 is well executed but it does no mention of how much their method improved emotion classification. The only result (that I saw) was the result of their method with no baseline comparison.

**Questions:**

Why is it claimed that accuracies in a dissimilarity matrix map represents how emotions are encoded in LLMs? Can you please make this argument clearer?

---

> ### Author Response · Authors · 2025-11-22
> **Response to reviewer kBVa**
>
> We thank the reviewer for the detailed feedback. We believe the revisions have significantly strengthened the paper and aligned its claims more closely with the evidence we present/avoid overclaiming.
>
> > By asking the LLM to classify the emotion of the text, there is the worry that you are “priming” the network to focus on emotion...
>
> We aim to study how the model organizes representations when tasked with affective classification, rather than to model all possible ways the model might internally process emotion during arbitrary next-token prediction. Sec. 3.2 now more clearly states that we are analyzing the geometry of representations in an explicit emotion-classification setting, in line with affective computing protocols. We explicitly list this as a limitation in the Discussion: extending the analysis to more “naturalistic” settings is an important direction for future work, but beyond the current study.
>
> >The paper examines hidden states but not effects of each sub-layer
>
> We agree that decomposing layers into sub-components would be informative. In this work we deliberately focus on the mean-pooled full layer outputs (Sec. 3.2), which are the standard objects used in major probing studies. We clarify in Sec. 3.2 that the activations we analyze are the layer outputs after pooling and have added this as a limitation in the Discussion.
>
> >Results are only on two LLMs
>
> We have addressed this by adding a new set of experiments on LLaMA-3-70B-Instruct (Sec. 7). These new results strengthen the case that the observed affective geometry does indeed generalize to a larger/instruction-tuned model.
>
> >GoEmotions is on reddit dataset which has one mode of writing…
>
> We highlight this constraint more explicitly in the limitations section.
>
> > Figure 2 appears to show the geometry of the MDS accuracy-space, not the intrinsic emotional geometry of the LLM’s hidden states… representing each emotion as a single point…
>
> We agree that our main figures visualize the geometry of pairwise separability, induced by logistic probes, rather than raw hidden states. This design was deliberate, but we have clarified the purpose (to dovetail with our uncertainty quantification) and added related robustness checks via cosine similarity in Apx. A.
>
> > It is not clear how VAD scores were produced for the GoEmotions dataset.
>
> Sec. 4.3 now makes the procedure more explicit, and we've modified Figure 2 to also reflect the ANEW VAD scores explicitly.
>
> >The comparison with EEG data is again a comparison logistic regression accuracy scores… I am not sure this shows any mapping between how the two types of data represent emotions.
>
> We agree that the EEG analysis is based on separability geometry (pairwise logistic regressions) rather than raw feature spaces, and we have clarified our claims accordingly. In Sec. 4.4, we clarify that we are comparing the latent structure of the emotion landscape across LLM activations and EEG features; we do not claim a direct representational isomorphism. In the Limitations, we explicitly state that the observed parallels should be interpreted as a structural analogy; this does not constitute evidence of shared cognitive encoding or one-to-one mapping between LLM and neural codes.
>
> > from my understanding of reading Sec. 5, the geometric structure analyzed is of the separability matrix, not of the hidden-states themselves.
>
> Correct. The new cosine-distance ablation (Apx. A) directly addresses the concern that this geometry might be purely probe-induced by showing that semantic structure emerges when the distance matrix is built from hidden-state cosine dissimilarities without any logistic regression. Our goal is to study the geometry of how emotions are arranged relative to one another in representation space, as reflected in separability. We now make this more explicit throughout.
>
> >Section 6 is well executed but it does no mention of how much their method improved emotion classification.
>
> We have now edited Section 6 to make the baseline classification (and resulting improvement) explicit.
>
> > Why is it claimed that accuracies in a dissimilarity matrix represents how emotions are encoded in LLMs? Can you please make this argument clearer?
>
> Conceptually, if two emotion categories are encoded similarly in the hidden space (their activation distributions overlap heavily), any linear classifier will have difficulty separating them, yielding low logistic accuracy for that pair (and vice versa for diff encodings). Thus, the matrix of pairwise accuracies provides a data-driven description of relative cluster separability in activation space.
>
> In the revision, Sec. 3.1 now explicitly states this intuition; we also motivate the pairwise dissimilarity matrix via linking it to our uncertainty quantification application. The new cos-dist experiment (Apx. A) further supports that the separability geometry we observe is consistent with the geometry of mean hidden states themselves.

---

> > ### Comment · Reviewer_kBVa · 2025-11-26
> >
> > Even with the updates made, the limitations of the study pointed out before outweigh the updates.
> > Testing only on a single dataset with one mode of writing is just not enough of an analysis.
> > Basing the analysis on MDS space which is derived from the dissimilarity matrix (a matrix of accuracy scores) does not convince me that the findings reflect how emotions are represented by LLMs. At most this type of analysis can show which emotions may co-activate and which ones do not. This may allowing binning emotions together and pushing them apart. However, I am not sure this type of analysis can allow you to make the claim that emotional representations are linear vs non-linear, the rank of the representation, the curvature, etc. I still think you are overclaiming given the analysis done.
> >
> > Additionally, still not much details were given for how VAD scores were derived and from the sound of it VAD scores were given for each "lexical item". Does that mean each word was given a VAD score which was then somehow merged across the sentence? How is that done? Is this reliable? Does it work under negation?
> >
> > "we clarify that we are comparing the latent structure of the emotion landscape across LLM activations and EEG features"
> > Again, at most I think you can claim that similar emotions co-activate in LLMs and EEG features. This again allows binning emotions together and pushing them apart. I am not sure that you can claim to compare the latent structure of LLM activation and EEG features given the analysis done.

---

> ### Author Response · Authors · 2025-11-26
> **Response to reviewer kVBa**
>
> We thank the reviewer for the follow-up comments and for engaging with the revised manuscript. We would like to respectfully clarify what appears to be a misunderstanding of our experimental setup—particularly around how VAD scores are constructed—which we believe may underlie several of the reviewer’s concerns (e.g., “co-activation,” lexical aggregation, and the use of a single dataset). We are of course happy to answer any follow-up questions.
> >1. Clarifying how VAD scores are used (no word-level scoring or aggregation)
> The reviewer’s question suggests that VAD values may have been assigned per lexical item and then aggregated across a sentence, and that this may affect interpretability (e.g., under negation). We want to clarify that our method does not involve any per-word scoring, token-level scoring, or aggregation across words.
>
> Specifically:
> - Each GoEmotions example is a single labeled text span (corroborated by multiple independent human raters).
> GoEmotions provides categorical emotion labels at the utterance level.
> - Each categorical label is mapped to a single VAD vector from the ANEW norms. Thus, each training example inherits a VAD vector directly from its emotion label, not from its tokens.
> - No step assigns VAD values to words, and no averaging or merging across lexical items occurs.
> - We believe this misunderstanding may also underlie the “co-activation” interpretation. Our analysis does not track activations of individual words; instead, it evaluates the separability of the hidden-state distributions associated with the categorical labels of entire supervised examples.
>
> > 2. Clarifying the single-dataset limitation in light of the above
>
> The reviewer expresses concern that using a single dataset with “one mode of writing” may weaken the conclusions. We agree this is a limitation and state so in the paper. However, we also want to clarify that the concern may stem from the assumption that VAD scores were computed from lexical content—if so, the writing style of individual sentences would indeed matter much more.
>
> In our actual setup, however:
> - The writing style of the sentence does not affect VAD assignment.
> - All VAD values come from the emotion label, not the sentence content. Therefore, the dataset’s limitation w.r.t. emotion representation is primarily about breadth of emotion categories, not lexical variability.
> - To our knowledge, GoEmotions is currently the only publicly available dataset that simultaneously provides:
>   - A large number of emotion categories (e.g., multiple classes for positive/negative/ambiguous)
>   - Sufficient sample counts per category to support probing, and
>   - Emotion labels that can be coherently mapped onto existing affective-norm resources like ANEW.
> Thus, the reliance on a single dataset is a practical constraint rather than a methodological choice driven by lexical aggregation.
>
> >3. Summary
>
> We appreciate the reviewer’s thoughtful engagement. Our primary goal here is to clarify that:
> - We do not derive VAD scores from words or perform lexical aggregation.
> - Each example’s VAD value is inherited directly from its emotion label.
> - Several raised concerns—including those related to “co-activation” and dataset limitations—appear to be based on an interpretation of the setup that we do not use.
>
> We hope this clarification helps contextualize the presented analysis, and we would be very happy to answer any further questions

---

### Official Review · Reviewer_LxRs · 2025-11-01

**Soundness:** 3
**Presentation:** 3
**Contribution:** 3
**Rating:** 2
**Confidence:** 5

**Summary:**

This work tries to answer whether LLMs (Gemma-2-9B and Mistral-7B) actually encode emotion structures similar to human models of emotion, The authors use a technique using pairwise logistic probing on the GoEmotions dataset, visualize the latent space withgeometric embedding (MDS/Isomap), and statistical alignment (Procrustes analysis) with human data statistically map this geometry onto well-known human psychological models (like ANEW valence arousal maps) using Procrustes analysis, including EEG emotion manifolds. They cap this with a practical demonstration using geometric distance for quantifying uncertainty

**Strengths:**

- Original Concept: Using affective models as a benchmark for latent geometry is a really neat, original idea for interpretability research. It's meaningful. Using the well-established structure of human affect (valence/arousal) as a geometry benchmark for LLM interpretability is clever and novel.

- Cross-Modal Insight: The comparison to human EEG data is a creative touch that makes the structural alignment claims much more interesting. The analysis uses appropriate stats like permutation tests and trustworthiness metrics, that's good rigor.

- Interdisciplinary: Comparing LLM structures to EEG emotion manifolds is a creative and refreshing cross-domain perspective.

- Practical Utility: The section on uncertainty quantification is a thoughtful application that proves these geometric insights have promise for safety applications.

- Clear Writing: The paper is easy to follow, and the experimental flow makes sense. The combination of probing, embedding, and Procrustes alignment is technically sound and robust. The use of permutation tests for significance is a strong point.

**Weaknesses:**

This is a paper with novel methodology that will encourage discussion by linking affective science and LLM geometry. The methodology is strong, but the authors must address the limitations around linear bias, small model scope, and the measurement of manifold of the human-LLM parallels. and the geometric analysis could be expanded, the study provides genuine insight and is methodologically clear. it's likely to generate good discussion.


- Heavy on Linear Probing: Relying only on logistic regression assumes the emotion categories are linearly separable. This might really bias the geometry they end up seeing. It would be stronger to include a non-linear probe (like a kernel method or maybe UMAP embeddings) to check this. to truly validate the latent manifold claims

- Limited Demonstration (evaluation): They only tested two open-source small (7B–9B) models, and they are both relatively small. We don't know if these results will hold up for larger, instruction-tuned, or multimodal systems. Generalizability is a concern.

- Interpretational Overreach: The alignment with EEG data is a correlation and should be framed more cautiously. Structural alignment is not evidence of cognitive isomorphism; this point needs to be softened in the text. Also, The EEG alignment is interesting, but the claim needs check or proof. it's more of an analogy of shared structure, not firm evidence of shared cognitive encoding.

- "Manifold" Claims are Vague: The eigenspectra and Isomap results actually suggest the structure is fairly diffuse and high-rank. Calling this a clean "affective manifold" needs explanation. This papers own eigenspectrum and Isomap results point to a diffuse, high-rank structure, which works against the central idea of a clean, low-dimensional affective manifold. This internal contradiction should be clarified

- Citation needed during discussion of Background and when motivating that distributed embeddings capture affective dimensions. Prior work showing how affect (valence/arousal/dominance) maps onto word embeddings and methods to retrofit standard embeddings to capture affective dimensions. the paper’s claim that latent spaces capture valence–arousal.  Shah, S., Reddy, S., & Bhattacharyya, P. (2022). Affective Retrofitted Word Embeddings (AACL 2022) ACL Anthology


Suggestions

- should ablate or test sensitivity to other alternative distance metrics (e.g., Mahalanobis or just plain cosine) to ensure the geometry isn't picked by author.

- Add a clearer, quantitative summary of the Procrustes alignment quality (like a scatter or correlation plot). Include a simple scatter/correlation plot showing the LLM space vs. the human space to visually assess the Procrustes quality.

- Make the language more clear when discussing human-LLM parallels; focus on structural vs. cognitive similarity. emphasize structural analogy rather than shared representations

- A measurement report with curvature would better support (or refute) the "manifold" interpretation.

**Questions:**

Please see the concerns raised in the Weaknesses section

---

> ### Author Response · Authors · 2025-11-22
> **Response to reviewer LxRs**
>
> We thank the reviewer for the detailed read of our submission and for the helpful feedback.
>
> >  The methodology is strong, but the authors must address the limitations around linear bias, small model scope, and the measurement of manifold of the human-LLM parallels. and the geometric analysis could be expanded, the study provides genuine insight and is methodologically clear. it's likely to generate good discussion.
>
> We believe our approach introduces a novel lens on LLM representations and are glad to hear the encouraging feedback on the methodology. The limitations that you list all present very interesting directions for future work. Notably, we have conducted several additional experiments, which begin to address some of the limitations and have expanded our discussion section in the hope that some of these points can motivate future research within the community.
>
> > Heavy on Linear Probing
>
> We agree that it is important to ensure the geometry is not biased by arbitrary methodological choices. To address this, we added UMAP embeddings (Appendix A), which recover the same global structure, including the valence–arousal axis and parabolic shape, in addition to corroborating the structure outside of any logistic regression via activation cosine similarity analysis (see Comment on cosine similarity below). This helps corroborate that the observed geometry is not induced by the logistic probe but reflects structure already present in the raw hidden-state activations.
>
> > Limited Demonstration (evaluation): They only tested two open-source small (7B–9B) models, and they are both relatively small. We don't know if these results will hold up for larger, instruction-tuned, or multimodal systems. Generalizability is a concern.
>
> We addressed this by repeating all primary analyses on LLaMA-3-70B-Instruct, a substantially larger and instruction-tuned model. We reproduce the full set of findings, showing that our conclusions hold for a much larger model.
>
> > Interpretational Overreach: The alignment with EEG data is a correlation and should be framed more cautiously.
>
> We agree that the similarities in latent structure that we observed in our analysis do not amount to evidence of shared cognitive encoding. We have made several edits in the manuscript, including in the limitations section, to clarify this/frame more cautiously.
>
> > "Manifold" Claims are Vague
>
> We have clarified the manuscript to distinguish between (i) the overall high-rank structure of the distance matrix and (ii) the presence of meaningful low-dimensional directions that deviate from this high-rank bulk. As noted in the text (Sec. 5), the diffuse eigenspectrum is expected: most variance reflects noise and task-irrelevant structure, while affective geometry appears only as statistically significant components that separate from the bulk. Accordingly, we are careful to not claim the existence of a clean, globally low-dimensional “affective manifold,” but rather explore the emergence of axes/eigengaps above the bulk spectrum. This resolves the apparent tension between diffuse spectra and the presence of interpretable affective structure.
> > Citation needed during discussion of Background and when motivating that distributed embeddings capture affective dimensions.
>
> We have now added a citation to \citet{shah2022affective}.
>
> > should ablate or test sensitivity to other alternative distance metrics (e.g., Mahalanobis or just plain cosine) to ensure the geometry isn't picked by author.
>
> We agree that the geometry should be robust to the choice of distance metric. To test this, we added an ablation using a probe-free cosine-distance matrix computed directly from the mean hidden-state activations for each emotion. Results (see Appendix A) are corroborated under this alternative formulation, indicating that the observed geometry does not depend on the specific distance metric used and is a stable property of the underlying activations.
>
> > add a clearer, quantitative summary of the Procrustes alignment quality (like a scatter or correlation plot).
>
> We thank the reviewer for this suggestion. We have now added a clearer quantitative summary of the Procrustes alignment, including the requested scatter and correlation plots comparing the LLM embedding to the ANEW valence–arousal coordinates.
> > Make the language more clear when discussing human-LLM parallels; focus on structural vs. cognitive similarity. emphasize structural analogy rather than shared representations
>
> We have edited the manuscript accordingly.
> > A measurement report with curvature would better support (or refute) the "manifold" interpretation.
>
> This is an interesting suggestion. Estimating curvature on high-dimensional data empirically is a challenging problem in geometric data analysis. While it would be interesting to test curvature estimators in this setting, such an extension merits a separate study and is unfortunately beyond the scope of the present paper.

---

### Official Review · Reviewer_FbMh · 2025-11-01

**Soundness:** 3
**Presentation:** 3
**Contribution:** 3
**Rating:** 6
**Confidence:** 3

**Summary:**

This paper investigates how large language models (LLMs) internally represent emotions, using the GoEmotions dataset as a testbed. By analyzing activations from Gemma-2-9B and Mistral-7B, the authors find that emotion representations in these models align with the well-known valence-arousal model from psychology and exhibit a parabolic shaped geometry resembling human affective structure. The study compares these LLM embeddings to human EEG data, showing similar geometric layouts, suggesting parallels between artificial and biological emotion processing. Using linear (MDS) and nonlinear (Isomap) manifold analyses, the authors conclude that affective representations are mostly linear with mild nonlinearity, consistent with the linear representation hypothesis. Finally, they demonstrate that geometric distances in this space can be used to quantify uncertainty in emotion classification, yielding well-calibrated confidence estimates.

**Strengths:**

- The paper presents a creative, interdisciplinary bridge between computational neuroscience and LLM interpretability, linking artificial and human affective structures.
- The paper provides quantitative and visual evidence that LLMs encode emotion categories in ways consistent with psychological valence-arousal models and EEG data.
- The discovery that LLMs exhibit modestly nonlinear yet largely linear affective geometry helps reconcile competing interpretability hypotheses.
- Figures effectively visualize emotion clusters, parabolic structures, and the connection between emotion axes and valence.

**Weaknesses:**

- The study is limited to two LLMs (Gemma-2-9B, Mistral-7B); generalization to larger or instruction-tuned models remains untested.
- The datasets (GoEmotions, FACED) are small and culturally specific, possibly limiting the universality of the findings.
- The causal link between latent geometry and model behavior (e.g., emotion reasoning, empathy) is not established.

**Questions:**

- How stable is the observed valence–arousal alignment across model families (e.g., LLaMA, GPT, Claude)?
- Could fine-tuning on emotion datasets distort or strengthen the parabolic geometry found here?
- Could the geometric features observed be used to steer model emotional tone or improve alignment for safety applications?

---

> ### Author Response · Authors · 2025-11-22
> **Response to reviewer FbMh**
>
> We thank the reviewer for their careful read of our submission, the helpful feedback and the interesting questions.
>
> > The study is limited to two LLMs (Gemma-2-9B, Mistral-7B); generalization to larger or instruction-tuned models remains untested.
>
> We appreciate this important point. In response, we conducted a full slate of experiments using LLaMA-3-70B-Instruct, a substantially larger and instruction-tuned model. We repeated all main analyses (pairwise logistic probing, CMDS, ISOMAP, Procrustes alignment, and intrinsic-dimension estimation) without changing methodology or hyperparameters.
> Across all analyses, the valence–arousal parabolic geometry persists and becomes even clearer in the larger model. These results help strengthen the generality of our findings and are now included in Section 7 and Appendix A.
>
> > The datasets (GoEmotions, FACED) are small and culturally specific, possibly limiting the universality of the findings.
>
> Thank you for the comment. We agree that this is a limitation, which we also discuss in the limitations section. An analysis of a wider range of data sets is unfortunately beyond the scope of the current study, but we have expanded the limitations section to emphasize this point.
>
> > The causal link between latent geometry and model behavior (e.g., emotion reasoning, empathy) is not established.
>
> Thank you for the comment. In the revised manuscript, we emphasize that we observe structural similarity, but that this does not constitute evidence of shared cognitive encoding. We have reframed the claims accordingly throughout the manuscript.
>
> > How stable is the observed valence–arousal alignment across model families (e.g., LLaMA, GPT, Claude)?
>
> To assess stability beyond our original models, we repeated all analyses on LLaMA-3-70B-Instruct. The valence–arousal alignment remains robust: 34 layers show significant 2D Procrustes alignment with ANEW (p₂D < 0.05), and the embeddings exhibit the same characteristic parabolic structure observed in Gemma and Mistral. Extending this evaluation to GPT or Claude models poses challenges due to their proprietary nature, but we note an extension to other language families as a direction for future work.
>
>
> > Could fine-tuning on emotion datasets distort or strengthen the parabolic geometry found here?
>
> This is an interesting question. While dedicated emotion-specific fine-tuning experiments are beyond the scope of the present paper, we note that our new results using an instruction-tuned model (LLaMA-3-70B-Instruct) indicate that the affective geometry remains stable even under substantial instruction tuning. We highlight a systematic study of emotion-focused fine-tuning as an important direction for future work.
>
> > Could the geometric features observed be used to steer model emotional tone or improve alignment for safety applications?
>
> Thank you for the suggestion. Indeed, we believe that this work may motivate future safety applications. However, the present paper focuses on an exploratory geometric analysis; the development of applications merits a separate study. We have expanded our discussion of future work accordingly.

---

### Author Response · Authors · 2025-11-22
**Response to all reviewers**

We would like to thank all reviewers for carefully reading our manuscript and for the detailed comments.

In addressing the reviewers’ feedback, we have made the following main changes to the manuscript:

>1. Generalization to larger / instruction-tuned LLMs.

We have incorporated a new slate of experiments on LLaMA-3-70B-Instruct into all main analyses (pairwise logistic probing, classical MDS, ISOMAP, and alignment statistics). Across all evaluations, we observe that our main findings—including the emergent valence-arousal structure—persist in LLaMA-3-70B-Instruct.

>2. Covariance-based distances replicate the same geometry.

To address concerns that the geometry might be an artifact of logistic-regression decision boundaries,
we computed an alternative distance matrix using the cosine dissimilarity between the mean hidden-state activations
for each emotion, defined as $d_{\text{cos}}(i,j) = 1 - \cos(\bar{h}_i, \bar{h}_j)$, where $\bar{h}_i$ is the average
activation vector for emotion $i$. The resulting layout of emotions preserves the semantically coherent
valence--arousal organization (as seen in Appendix A and Fig. 13), indicating that the geometry is not dependent on the logistic
regression model. We also clarify in the main text (line 133) that logistic probing remains useful because it provides
a well-defined separating hyperplane, which is required for our uncertainty-quantification analysis.

>3. Added nonlinear embedding analysis (UMAP).

We added UMAP embeddings (Appendix A). Their clusters and dominant axes align with the ANEW valence-arousal dimensions, confirming that the reported structure is not an MDS artifact.

>4. ANEW valence-arousal comparison plot added.

We now include a direct visualization of the ANEW valence--arousal space alongside our LLM activation MDS embeddings (Fig.2), making the alignment clearer.

>5. Clearer and more cautious framing of both LLM-EEG comparisons and linear vs. manifold structure.

Following reviewer guidance, we revised the manuscript to emphasize that we observe structural similarity rather
than evidence of shared cognitive encoding (Section 4.4). We now explicitly frame all LLM-EEG comparisons as an
analogy (e.g., as opposed to a biological claim), and clarify this throughout the Discussion and Limitations sections. We also
refined our language regarding linear versus manifold structure (Sections 3 and 5) to provide a more consistent and
appropriately cautious framing of these results.

We believe that the reviewers’ feedback and the resulting revisions have significantly strengthened the manuscript.

We provide detailed responses to each reviewer in separate comments below.

---

### Meta-Review · Area_Chair_Ksqn · 2026-01-13

**Summary:**

This submission studies the geometry of LLM representations from a psychological perspective, using emotions as a test bed. The results are empirical and based on the GoEmotions corpus. They find that LLMs indeed represent emotions coherently, provide empirical evidence for the linear representation hypothesis, and quantify uncertainty in emotion processing
tasks.

A common concern from reviewers from the limited evaluation on only two models, which was increased to three during the rebuttal, as well as only one dataset. Also, several concerns regarding the conclusions and methods were raised. While overall reviewers appreciated the problem and idea, the execution is not convincing.

**Reviewer Concerns:**

- FbMh Only two LLMs: Yes, but concern still remains (only one model)
- FbMh Small datasets: **No**
- FbMh Causal link: **No**
- LxRs Heavy on Linear Probing: **No**
- LxRs Limited Demonstration: Yes, but concern still remains (only one model)
- LxRs Interpretational Overreach: Yes
- LxRs "Manifold" Claims: Yes
- LxRs Citations: Yes
- kBVa “priming” the network: **No**
- kBVa effects of each sub-layer: **No**
- kBVa Only two LLMs: **No**
- kBVa reddit dataset: **No**
- kBVa Figure 2: **No**
- kBVa VAD scores: **No**
- kBVa comparison with EEG data: **No**
- kBVa separability matrix: **No**
- kBVa Section 6: **No**
- kBVa how emotions are encoded in LLMs: **No**
- c3Kz ISOMAP: Yes

**Reviewer Scores:**

FbMh 6->6
LxRs 2->4
kBVa 2->2
c3Kz 8->8

---

### Decision · Program_Chairs · 2026-01-26

Reject